# Invasive Diagnostic and Therapeutic Management of Cerebral VasoSpasm after Aneurysmal Subarachnoid Hemorrhage (IMCVS)—A Phase 2 Randomized Controlled Trial

**DOI:** 10.3390/jcm11206197

**Published:** 2022-10-20

**Authors:** Hartmut Vatter, Erdem Güresir, Ralph König, Gregor Durner, Rolf Kalff, Patrick Schuss, Thomas E. Mayer, Jürgen Konczalla, Elke Hattingen, Volker Seifert, Joachim Berkefeld

**Affiliations:** 1Department of Neurosurgery, University Hospital Bonn, 53127 Bonn, Germany; 2Department of Neurosurgery, University of Ulm, Günzburg, 89081 Ulm, Germany; 3Department of Neurosurgery, Jena University Hospital, 07743 Jena, Germany; 4Department of Neuroradiology, Jena University Hospital, 07743 Jena, Germany; 5Department of Neurosurgery, University Hospital Frankfurt, 60590 Frankfurt am Main, Germany; 6Department of Neuroradiology, University Hospital Frankfurt, 60590 Frankfurt am Main, Germany

**Keywords:** intracranial aneurysm, treatment, subarachnoid hemorrhage, vasospasm, delayed ischemic neurological deficit, balloon angioplasty, intra-arterial spasmolysis

## Abstract

Cerebral vasospasm (CVS) is associated with delayed cerebral ischemia (DCI) after aneurysmal subarachnoid hemorrhage (SAH). The most frequently used form of rescue therapy for CVS is invasive endovascular therapy. Due to a lack of prospective data, we performed a prospective randomized multicenter trial (NCT01400360). A total of 34 patients in three centers were randomized to invasive endovascular treatment or conservative therapy at diagnosis of relevant CVS onset. Imaging data was assessed by a neuroradiologist blinded for treatment allocation. Primary outcome measure was development of DCI. Secondary endpoints included clinical outcome at 6 months after SAH. A total of 18 of the 34 patients were treated conservatively, and 16 patients were treated with invasive endovascular treatment for CVS. There was no statistical difference in the rate of cerebral infarctions either at initial or at the follow-up MRI between the groups. However, the outcome at 6 months was better in patients treated conservatively (mRs 2 ± 1.5 vs. 4 ± 1.8, *p* = 0.005). Invasive endovascular treatment for CVS does not lead to a lower rate of DCI but might lead to poorer outcomes compared to induced hypertension. The potential benefits of endovascular treatment for CVS need to be addressed in further studies, searching for a subgroup of patients who may benefit.

## 1. Introduction

Delayed cerebral ischemia (DCI) is an important cause of poor outcome after an aneurysmal subarachnoid hemorrhage (SAH) and is defined as a focal neurological impairment or a decrease of the patients’ Glasgow coma scale score as defined by Vergouwen et al [1,2]. Currently, orally administered Nimodipine is the only intervention to prevent DCI after SAH. However, this effect of Nimodipine seems to be due to neuroprotection rather than by vasodilation [3]. Cerebral vasospasm (CVS) is associated with DCI [4] that may develop by lowering cerebral blood supply below demand.

At present, medical therapy for CVS is induced hypertension [5]. The most frequently used form of rescue therapy for CVS is invasive endovascular therapy, including selective intra-arterial infusion of vasodilators and balloon angioplasty [6,7]. Due to a lack of prospective data for invasive diagnostic and the therapeutic management of CVS, we performed a prospective randomized study comparing DCI-related infarctions and the clinical outcome of patients with SAH and CVS with and without invasive endovascular treatment (NCT01400360).

## 2. Materials and Methods

Between August 2009 and September 2012, we performed a multicenter, randomized trial in 3 hospitals in Germany (University Hospital Frankfurt, University Hospital Ulm, and University Hospital Jena). The study was approved by the local ethics committee and all participating centers (Ethik-Kommission, University Hospital Frankfurt, ID 68/09). Written informed consent was obtained from all patients.

### 2.1. Definitions and Data Recording

SAH was diagnosed by computed tomography (CT) or lumbar puncture. All patients with spontaneous SAH underwent four-vessel digital subtraction angiography (DSA). Clinical data, including patient characteristics on admission and during treatment course, radiological features, and functional neurological outcome were collected and entered into a computerized database (SPSS Version 15 Institute, Inc, Chicago, IL, USA). Treatment decision (coiling versus clipping) was based on an interdisciplinary approach in each individual case. Aneurysm treatment was performed within 24 h after admission. Acute hydrocephalus was treated by external cerebrospinal fluid diversion. Osmotherapy and mild hyperventilation were used for the treatment of elevated ICP (>20 mmHg). Apart from close neurological monitoring, routine surveillance included daily transcranial Doppler measurements of red blood cell flow velocities. CT imaging was performed routinely (1) 24–48 h after aneurysm clip or coil obliteration to assess procedural complications, (2) on day 14–21 to diagnose delayed cerebral infarctions and to assess the necessity of a ventriculoperitoneal shunt, and (3) at variable time points whenever neurological deteriorations occurred. All patients underwent screening for CVS using MRI, including PWI/DWI, performed routinely on day 4–14 and cerebral angiography between day 7 and 10 [7]. Baseline MRI with PWI/DWI was also performed in any case of neurological deterioration of the patient or increased mean velocity ≥150 cm/s or an increase in velocity ≥50 cm/s within 24 h in transcranial Doppler sonography. Sufficient fluid was administered to maintain a high normal euvolemic status. All patients received nimodipine from the day of admission. In the case of hyponatremia fludrocortisone was added to the therapy.

### 2.2. Inclusion Criteria 

Patient age of 18–75 years. Patients who could undergo MRI scans initially and at follow-up. Patients with aneurysmal SAH WFNS grades I–IV with hemodynamically relevant CVS defined as: 1. Elevated time to peak (TTP) > 2 s compared to the corresponding contralateral side, or mean transit time (MTT) > 3.5 s. 2. Profound narrowing of cerebral vessels in MRA scan. 3. Existence of “tissue at risk” (vital brain tissue with DWI lesions <50%), as described before [8].

Patients were then randomized in one of the two groups, conservative versus invasive endovascular treatment, in a 1:1 allocation ratio. Patients without relevant CVS in DSA, patients with DWI lesions ≥ 50% of the relevant vessel territory, and patients who were not able to undergo MRI scans due to moribund clinical status or due to implants not safe for MRI imaging, as well as patients WFNS grade V, were excluded from the study.

### 2.3. Randomization

Patients were then randomized in one of the two groups, conservative versus invasive endovascular treatment, in a 1:1 allocation ratio (Figure 1). Patients without relevant CVS in DSA, patients with DWI lesions ≥ 50% of the relevant vessel territory, and patients who were not able to undergo MRI scans due to moribund clinical status or due to implants not safe for MRI imaging, as well as patients WFNS grade V, were excluded from the study.

### 2.4. Conservative Treatment

Arterial hypertension was induced with norepinephrine and fluids via central venous line. Mean arterial blood pressure (MAP) was raised to 110 mmHg. Induced hypertension was continued for 7 days. Thereafter, patients were reassessed for CVS using MRI. In patients with CVS, induced hypertension was continued for the following 7 days, and reassessment was performed thereafter, as described above. In patients with resolution of CVS, induced hypertension was terminated.

### 2.5. Invasive Endovascular Treatment

Whenever possible, proximal CVS was treated by transluminal balloon angioplasty (TBA) and distal or diffuse CVS was treated by intra-arterial application of nimodipine at the discretion of the treating neuroradiologist, as described previously [9,10].

### 2.6. Outcome Measurement and Endpoints

The flowchart of the diagnosis and treatment protocol, as well as the efficacy assessment of endovascular therapy, is published in Vatter et al. 2011 [8]. Efficacy of endovascular treatment was assessed by the treating neuroradiologist and graded into good versus fair. The effect of endovascular treatment was controlled 48 ± 12 h later with MRI. In case of DWI/PWI mismatch, the treatment cycle consisting of MRI, DSA (including endovascular treatment), and a follow-up MRI (48 ± 12 h later) was repeated until no further tissue at risk could be detected.

Imaging data for new infarctions during the phase of CVS was assessed by a neuroradiologist blinded for treatment allocation. The cerebrum was partitioned into 19 segments. A 50% DWI lesion ≥ 1 segment was defined as major and < 50% was defined as minor infarct. Only delayed spontaneous infarctions were counted as DCI-related cerebral infarctions. Lesions related to aneurysm treatment (clipping or coiling of the ruptured aneurysm) or caused by extraventricular drains, pre-existing infarcts, and hypodensities surrounding hematoma or in proximity to the site of surgery were excluded.

The primary outcome measure was the development of new DCI-related cerebral infarctions during the phase of CVS. Secondary endpoints included clinical outcome at 6 months after SAH according to the modified Rankin scale (mRs).

### 2.7. Sample Size Calculation

As 60–75% of SAH patients with relevant CVS (“tissue at risk”) develop DCI [11,12,13], the study was planned to decrease CVS related infarcts to 50% by invasive endovascular diagnostic and management, with α=0.05, and 80% power, for which 92 patients in a 1:1 randomization was needed. An interim analysis was planned after 34 patients.

### 2.8. Statistics

Data analyses were performed using the computer software package SPSS (IBM SPSS Statistics for Windows, Version 25.0. IBM Corp., Armonk, NY, USA). Unpaired *t*-test was used for parametric statistics. Categorical variables were analyzed in contingency tables using Fisher’s exact test. Results with *p* < 0.05 were considered statistically significant.

## 3. Results

### 3.1. Patient Characteristics

A total of 34 patients had CVS. Eighteen patients were treated conservatively, and sixteen patients were treated with invasive endovascular treatment for CVS (Table 1).

### 3.2. Initial MRI

Of the 18 patients treated conservatively, 5 patients had no DWI lesions, 10 patients had minor DWI lesions, and 3 patients had major DWI lesions in the initial MRI.

Of the 16 patients in the invasive endovascular group, 2 patients had no DWI lesions, 11 patients had minor DWI lesions, and 3 patients had major DWI lesions in the initial MRI.

The initial rate of no/minor DWI lesions compared to major DWI lesions did not differ between the assigned treatment groups (*p* = 1.0).

### 3.3. Follow-Up MRI

Of the 18 patients treated conservatively, 8 patients had no new DWI lesion, 7 patients had new minor DWI lesions, and 3 patients had major DWI lesions in the follow-up MRI.

Of the 16 patients in the invasive endovascular treatment group, 8 patients had no new DWI lesion, 4 patients had new minor DWI lesions, and 4 patients had major DWI lesions in the follow-up MRI.

The rate of no new/new minor DWI lesions compared to new major DWI lesions did not differ between the assigned treatment groups (*p* = 0.7).

### 3.4. Frequency, Success, and Complications of Endovascular Treatment

Of the 16 patients who underwent invasive endovascular treatment, 16 patients underwent one procedure, 11 patients underwent two procedures, 4 patients underwent three procedures, 2 patients underwent four procedures, and one patient underwent five procedures. Detailed data on frequency, success, and complications of invasive endovascular treatment are presented in Table 2 and Table 3.

### 3.5. Outcome

Clinical outcome after 6 months was better in patients treated conservatively compared to patients in the invasive endovascular treatment group (mRs 2 ± 1.5 vs. 4 ± 1.8, *p* = 0.004, Figure 2).

## 4. Discussion

In this randomized trial, patients in the conservative treatment group, i.e., treatment with induced hypertension after diagnosis of CVS, had a better clinical outcome compared to patients in the invasive endovascular group. Therefore, the study was stopped prematurely after randomization of 34 patients.

Induced hypertension with maintenance of euvolemia is currently used in order to perform hemodynamic augmentation [14] and increase cerebral blood flow as well as brain tissue oxygenation [15].

Suwatcharangkoon et al. [16] recently showed that failure to respond to induced hypertension in patients with symptomatic vasospasm leads to a higher rate of cerebral infarction and poor outcome after 1 year compared to the group of patients who respond to induced hypertension.

According to the guidelines for the management of SAH [14], induced hypertension is recommended in patients with symptomatic CVS. While the prematurely terminated study on Hypertension Induction in the Management of AneurysmaL subarachnoid haemorrhage with secondary IschaemiA (HIMALAIA) [17] did not support induced hypertension, it was underpowered to draw definitive conclusions. In a retrospective observational study, Haegens et al. [18] analyzed SAH patients with diagnosed CVS and compared DCI rates in the group of patients with induced hypertension and without. DCI rates were significantly lower in the induced hypertension group (20%) compared to the group without induced hypertension (33%). Furthermore, Haegens et al. concluded that the reduced DCI rates may also lead to a reduction of poor clinical outcome.

Patients with CVS at the ICU reflect a highly vulnerable subgroup of SAH patients who are at risk for DCI that may develop when cerebral blood supply decreases below demand. An invasive diagnostic and therapeutic management in patients with critically low cerebral blood supply may lead to varying blood pressure and decreased monitoring compared to the treatment at the ICU over a period of time, which may explain why the otherwise effective treatment of angiographic CVS does not translate into better outcomes [19].

Furthermore, CVS is only one explanation for poor outcome after SAH. It is possible that factors such as early brain injury or systemic illness may contribute to poor outcome.

In conclusion, invasive endovascular treatment seems to be effective in the treatment of visual CVS. However, despite its widespread use nowadays, there is no evidence that invasive endovascular treatment improves outcome in patients with CVS, whereas it seems to have a high rate of serious complications [20]. A possible benefit of invasive endovascular treatment on clinical outcome could be limited to a special subgroup of SAH patients with CVS; however, it is unclear which patient cohort this subgroup might be.

We therefore changed our treatment policy in patients with diagnosed CVS to induced hypertension only and published our prospective results in the subgroup of SAH patients meeting inclusion criteria of the IMCVS trial recently [21]. 

### Limitations

The most obvious limitation is the limited power due to the smaller study population size than planned. Therefore, the potential benefits of invasive diagnostic and endovascular treatment cannot be ruled out. However, the randomized controlled design and a clear definition of CVS are the strengths of the trial. Furthermore, the result that clinical outcome of patients in the group of invasive diagnostic and therapeutic management compared to the conservative group is not only similar, but significantly poorer, does not support the widespread use of endovascular treatment for CVS but necessitates further studies in order define a subgroup of SAH patients who may benefit from this kind of invasive treatment.

## 5. Conclusions

In conclusion, we found that invasive endovascular treatment in SAH patients with CVS does not lead to a lower rate of delayed cerebral ischemia or improved clinical outcome but might lead to poorer outcome compared to patients treated with induced hypertension for CVS. Potential benefits of endovascular treatment for CVS need to be addressed in further studies, searching for a subgroup of patients who may benefit.

## Figures and Tables

**Figure 1 jcm-11-06197-f001:**
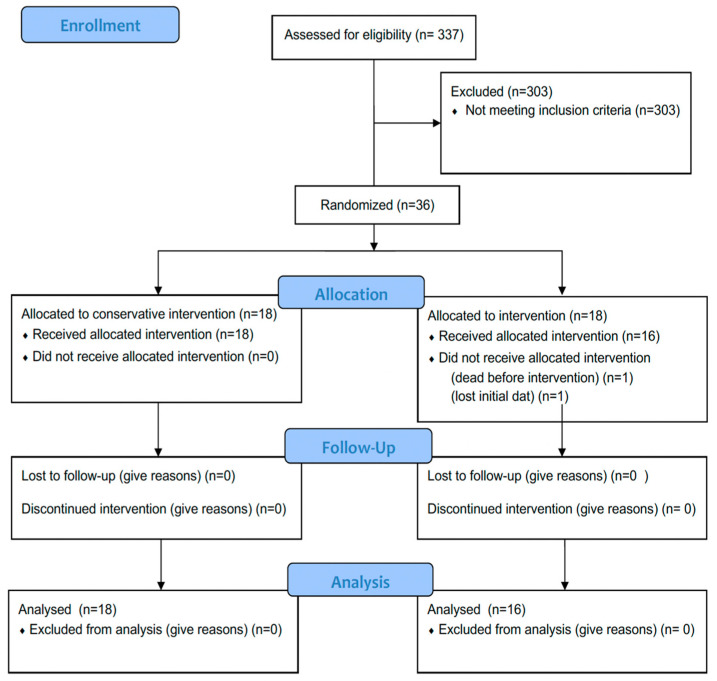
Diagram illustrating the flowchart regarding assessment, randomization, allocation, follow-up, and final analysis of study patients.

**Figure 2 jcm-11-06197-f002:**
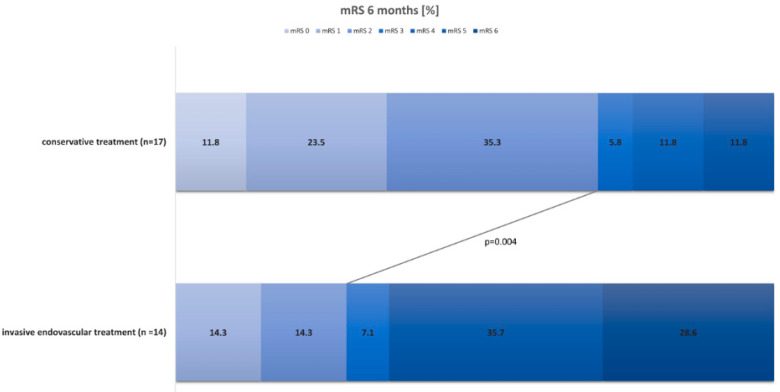
Column bars showing the proportions of patients regarding the mRS after 6 months. The first column bar represents the conservative treatment arm, whereas the second bar displays the invasive endovascular treatment arm. The line connects the mean mRS values of both groups and the *p*-value represents the result of the independent *t*-test.

**Table 1 jcm-11-06197-t001:** Patient characteristics.

Variable	Conservative (n = 18)	Invasive (n = 16)	*p*-Value
Age, Y ± SD, mean	55 ± 10	56 ± 12	
Smoker	9/14	9/14	1.0
Diabetes	0/18	2/15	0.2
Hypertension before SAH	8/18	8/15	0.7
Coronary heart disease	2/18	0/16	0.5
Adipositas	1/12	1/15	1.0
BMI	27 ± 3	22 ± 0.4	0.4
History of malignoma	0	0	
History of cerebral ischemia	1/18	1/16	
Extensive alcohol consumption	2/18	1/16	
History of myocardial infarction	2/18	0	
History of SAH (from the same aneurysm)	1/18	0	
mRS before SAH	0	0	
Karnofsky before SAH	100 ± 0	99 ± 3	0.1
Hydrocephalus at admission	13/16	12/16	1.0
GCS	9 ± 5	8 ± 4	0.6
WFNS grade	4 ± 2	4 ± 1	0.6
Fisher score	3 ± 0.5	3 ± 0.4	0.2
IVH	5/16	4/14	1.0
ICH < 3 cm	2/15	5/15	0.1
ICH > 3 cm	2/15	5/15	0.1
Clipping	10/18	8/18 *	
Coiling	8/18	10/18 *	

* number of patients assigned to invasive treatment. BMI = body mass index; GCS = Glasgow coma scale; ICH = intracerebral hemorrhage; IVH = intraventricular hemorrhage; mRS = modified Rankin scale; SAH = subarachnoid hemorrhage; Y = years.

**Table 2 jcm-11-06197-t002:** Invasive endovascular treatment.

Frequency of DSA	No. of pts.	Endovascular Treatment (i.a. Nimodipine/PTA)	No. of pts.	Treatment Success According to Interventionalist
1	16	i.a. nimodipine	16	Good (16/16)
2	11	i.a. nimodipine	7/11	Good (4/10)
				Fair (3/10)
		PTA	3/11	Good (3/10)
		none	1/11	Massive thromboembolic event
3	4	i.a. nimodipine	3/4	Good (1/4)
				Fair (1/4)
				No success (1/4)
		PTA	1/4	Good (1/4)
4	2	i.a. nimodipine	2	Good (0/2)
				Fair (1/2)
				No success (1/2)
5	1	i.a. nimodipine	1	Fair (1/1)

**Table 3 jcm-11-06197-t003:** Complications of invasive endovascular treatment.

Frequency of DSA	No. of pts.	Complications	No. of pts.	Severity of Complication/Necessity of Treatment
1	16	None		
2	11	Dissection	1	Stent implantation
		Bleeding	0	
		Thromboembolic event	1	Minor supratentorial infarction
		Embolic infarction	1	Massive cerebellar and brainstem infarction
3	4	Dissection	1	Stent implantation
		Bleeding	0	
		Thromboembolic event	1	Major supratentorial infarction
		Embolic infarction	0	
4	2	Dissection	0	
		Bleeding	0	
		Thromboembolic event	0	
		Embolic infarction	0	
5	1	None		

## Data Availability

All data are included in this manuscript.

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
