# Peer review of "Invasive Diagnostic and Therapeutic Management of Cerebral VasoSpasm after Aneurysmal Subarachnoid Hemorrhage (IMCVS)—A Phase 2 Randomized Controlled Trial"

_jcm, 2022, doi:10.3390/jcm11206197_

Round 1
Reviewer 1 Report
The authors describe a randomized controlled trial of the treatment of cerebral vasospasm in patients with aneurysmal SAH. The randomized 36 patients with aSAH and CVS based on radiographic criteria to endovascular treatment vs. conservative therapy. Ultimately the low sample size of this trial makes it too bias to draw any significant conclusions. The authors do not reference any previously described diagnostic criteria of CVS. The do not discuss the clinical status of patients in the diagnosis of CVS. How did they ensure there wasn't bias when only using radiographic criteria but then assessing clinical outcome. Average Karnofsky score was 99 in the invasive group - how is this possible when this is rated on a 10 point increment scale. The rate of endovascular complications is quite high which is what seems to drive the long term difference in mRS. Given the high rate of complications I am not sure this is generalizable.
Author Response
The authors describe a randomized controlled trial of the treatment of cerebral vasospasm in patients with aneurysmal SAH. The randomized 36 patients with aSAH and CVS based on radiographic criteria to endovascular treatment vs. conservative therapy. Ultimately the low sample size of this trial makes it too bias to draw any significant conclusions. The authors do not reference any previously described diagnostic criteria of CVS.
We agree with the reviewer, that one of the limitations of the study is its small sample size. However, it is the only prospective randomized study on this topic -till now.
Definitions of CVS and inclusion criteria for the study were described in a previous study (Vatter, H.; Güresir, E.; Berkefeld, J.; et al. Perfusion-diffusion mismatch in MRI to indicate endovascular treatment of cerebral vasospasm after subarachnoid haemorrhage. J Neurol Neurosurg Psychiatry 2011, 82, 876-883.).
In order to clarify, we changed “2.2. Inclusion criteria” (p.2):
Patients with aneurysmal SAH WFNS grades I-IV with hemodynamically relevant CVS defined as 1. Elevated time to peak (TTP) > 2 seconds compared to the corresponding contralateral side, or mean transit time (MTT) > 3.5 seconds. 2. Profound narrowing of cerebral vessels in MRA scan. 3. Existence of “tissue at risk” (vital brain tissue with DWI lesions <50%), as described before [8].
The do not discuss the clinical status of patients in the diagnosis of CVS. How did they ensure there wasn't bias when only using radiographic criteria but then assessing clinical outcome.
The primary endpoint of the study was development of new cerebral infarctions during the phase of CVS, as stated in “2.6. Outcome measurement and endpoints”.
The clinical status did not differ between the groups at presentation, as stated in Table 1.
Average Karnofsky score was 99 in the invasive group - how is this possible when this is rated on a 10 point increment scale.
15 of the 16 patients had a Karnofsky Score of 100, and one had a Score of 90. This resulted in a mean score of 99 ± 3. We decided to indicate mean values, in order to see these distinct differences, although not significant.
The rate of endovascular complications is quite high which is what seems to drive the long term difference in mRS. Given the high rate of complications I am not sure this is generalizable.
We agree, that the complication rates within the interventional group were higher compared to the conservative group. We are convinced that complications in this vulnerable patient population with significant vasospasm and “tissue at risk” occur during transport to, during, and after the intervention. Since we were keen to find a subgroup of patients who may benefit from endovascular intervention, we performed a prospective observational cohort trial and published it recently (Güresir et al 2022; https://doi.org/10.3390/jcm11195850).
We included the following into the manuscript (p.7): “We therefore changed our treatment policy in patients with diagnosed CVS to induced hypertension only and published our prospective results in the subgroup of SAH patients meeting inclusion criteria of the IMCVS trial recently [21].”
Reviewer 2 Report
Dear Authors,
I am glad to have the opportunity to review your work. This study aimed to assess the invasive diagnostic and therapeutic management of cerebral vasospasm after aneurysmal subarachnoid hemorrhage
The study is designed very well, with proper presentation of the results. My main concern is low data sample. However, the topic is great, and the results have clinical application and also address the need to explore the topic in the future.
I would suggest following changes:
1. In the Introduction section, add the definitions of delayed cerebral ischemia and cerebral vasospasm, from the literature.
2. In the methodology section, it needs to be more clear how you diagnosed DCI, using CT and/or MRI, and at which time frame. Also, you mention delayed cerebral infarctions in the Methodology. In the Introduction section, please add the definition on that also, and state a difference between diagnosis of DCI and delayed cerebral infarction. Also, in the results section, in the Tables, you presented several outcomes as infarctions – if they are delayed, please state them like that.

Author Response
Dear Authors,
I am glad to have the opportunity to review your work. This study aimed to assess the invasive diagnostic and therapeutic management of cerebral vasospasm after aneurysmal subarachnoid hemorrhage
The study is designed very well, with proper presentation of the results. My main concern is low data sample. However, the topic is great, and the results have clinical application and also address the need to explore the topic in the future.
I would suggest following changes:
- In the Introduction section, add the definitions of delayed cerebral ischemia and cerebral vasospasm, from the literature.
We included the definition of DCI according to Vergouwen et al 2010 as suggested (p.1):
“Delayed cerebral ischemia (DCI) is an important cause of poor outcome after aneurysmal subarachnoid hemorrhage (SAH), and is defined as focal neurological impairment or a decrease of the patients’ Glasgow coma scale score as defined by Vergouwen et al. [1, 2]”
Furthermore, we changed the “2.2. Inclusion criteria”, in order to clarify (p.2):
Patients with aneurysmal SAH WFNS grades I-IV with hemodynamically relevant CVS defined as 1. Elevated time to peak (TTP) > 2 seconds compared to the corresponding contralateral side, or mean transit time (MTT) > 3.5 seconds. 2. Profound narrowing of cerebral vessels in MRA scan. 3. Existence of “tissue at risk” (vital brain tissue with DWI lesions <50%), as described before [8].
- In the methodology section, it needs to be more clear how you diagnosed DCI, using CT and/or MRI, and at which time frame. Also, you mention delayed cerebral infarctions in the Methodology. In the Introduction section, please add the definition on that also, and state a difference between diagnosis of DCI and delayed cerebral infarction. Also, in the results section, in the Tables, you presented several outcomes as infarctions – if they are delayed, please state them like that.
We modified “2.6. Outcome measurement and endpoints”, and included (p.4):
“The flowchart of the diagnosis and treatment protocol, as well as the efficacy assessment of endovascular therapy is published in Vatter et al. 2011 [8]. Efficacy of endovascular treatment was assessed by the treating neuroradiologist and graded into good versus fair. The effect of endovascular treatment was controlled 48 ± 12h later with MRI. In case of DWI/PWI mismatch, the treatment cycle consisting of MRI, DSA (including endovascular treatment) and a follow-up MRI (48 ± 12h later) was repeated, until no further tissue at risk could be detected.
Imaging data for new infarctions during the phase of CVS was assessed by a neuroradiologist blinded for treatment allocation. The cerebrum was partitioned into 19 segments. A 50% DWI lesion 1 segment was defined as major, and < 50% was defined as minor infarct. Only delayed spontaneous infarctions were counted as DCI related cerebral infarctions. Lesions related to aneurysm treatment (clipping or coiling of the ruptured aneurysm) or caused by extraventricular drains, preexisting infarcts, and hypodensities surrounding hematoma or in proximity to the site of surgery were excluded.
The primary outcome measure was the development of new DCI related cerebral infarctions during the phase of CVS. Secondary endpoints included clinical outcome at 6 months after SAH according to the modified Rankin scale (mRs).”
We included the definition of DCI according to Vergouwen et al (2010) in the introduction section:
“Delayed cerebral ischemia (DCI) is an important cause of poor outcome after aneurysmal subarachnoid hemorrhage (SAH), and is defined as focal neurological impairment or a decrease of the patients’ Glasgow coma scale score as defined by Vergouwen et al. [1, 2].”
Furthermore, we modified the section 2.2. (p.2) in order to clarify the inclusion criteria and imaging modality (MRI).
“Patients with aneurysmal SAH WFNS grades I-IV with hemodynamically relevant CVS defined as 1. Elevated time to peak (TTP) > 2 seconds compared to the corresponding contralateral side, or mean transit time (MTT) > 3.5 seconds. 2. Profound narrowing of cerebral vessels in MRA scan. 3. Existence of “tissue at risk” (vital brain tissue with DWI lesions <50%), as described before [8].”
Cerebral ischemia in the Tables is “History of cerebral ischemia” (Table 1), and the result of complications in Table 3.
Reviewer 3 Report
Vatter et al report their results of their RCT evaluating the efficacy of endovascular therapy as compared to conservative management of cerebral vasospasm after aneurysmal subarachnoid hemorrhage between 2009 and 2012.
The negative result of the intervention is interesting in that it seems not only was there not a significant advantage but harm to endovascular provision of IA nimodipine. This negative trial answers the question of the utility of IA nimodipine and corroborates somewhat the outcome observed by Dr. Loch McDonald’s group using nimodipine nanoparticles through the EVD as intraventricular treatment after aneurysmal SAH.
This manuscript has taken nearly 10 years to come to reviewers and would be interesting to know, as the numbers of patients in the study are small, and if possible, the long-term outcome of these patients.
Otherwise, it was a well thought out negative study that is important to be available in published form.
Author Response
Vatter et al report their results of their RCT evaluating the efficacy of endovascular therapy as compared to conservative management of cerebral vasospasm after aneurysmal subarachnoid hemorrhage between 2009 and 2012.
The negative result of the intervention is interesting in that it seems not only was there not a significant advantage but harm to endovascular provision of IA nimodipine. This negative trial answers the question of the utility of IA nimodipine and corroborates somewhat the outcome observed by Dr. Loch McDonald’s group using nimodipine nanoparticles through the EVD as intraventricular treatment after aneurysmal SAH.
This manuscript has taken nearly 10 years to come to reviewers and would be interesting to know, as the numbers of patients in the study are small, and if possible, the long-term outcome of these patients.
Otherwise, it was a well thought out negative study that is important to be available in published form.
We would like to thank the reviewer for his kind response. Unfortunately, we only have outcome data at 6 months.